# Increased FOXM1 Expression by Cisplatin Inhibits Paclitaxel-Related Apoptosis in Cisplatin-Resistant Human Oral Squamous Cell Carcinoma (OSCC) Cell Lines

**DOI:** 10.3390/ijms21238897

**Published:** 2020-11-24

**Authors:** Hyeong Sim Choi, Young-Kyun Kim, Kyung-Gyun Hwang, Pil-Young Yun

**Affiliations:** 1Department of Oral and Maxillofacial Surgery, Section of Dentistry, Seoul National University Bundang Hospital, 82 Gumi-ro 173 beon-gil, Bundang-gu, Seongnam-si, Gyeonggi-do 13620, Korea; r2013@snubh.org (H.S.C.); kyk0505@snubh.org (Y.-K.K.); 2Department of Dentistry and Dental Research Institute, School of Dentistry, Seoul National University, 101 Daehak-ro, Jongno-gu, Seoul 03080, Korea; 3Division of Oral and Maxillofacial Surgery, Department of Dentistry, College of Medicine, Hanyang University, 222-1 Wangshimniro, Seongdong-gu, Seoul 04763, Korea; hkg@hanyang.ac.kr

**Keywords:** OSCC, cisplatin, paclitaxel, FOXM1, apoptosis

## Abstract

Cisplatin and paclitaxel are commonly used to treat oral cancer, but their use is often limited because of acquired drug resistance. Here, we tested the effects of combined cisplatin and paclitaxel on three parental (YD-8, YD-9, and YD-38) and three cisplatin-resistant (YD-8/CIS, YD-9/CIS, and YD-38/CIS) oral squamous cell carcinoma (OSCC) cell lines using cell proliferation assays and combination index analysis. We detected forkhead box protein M1 (FOXM1) mRNA and protein expression via real-time qPCR and Western blot assays. Cell death of the cisplatin-resistant cell lines in response to these drugs with or without a FOXM1 inhibitor (forkhead domain inhibitory compound 6) was then measured by propidium iodide staining and TdT dUTP nick end labeling (TUNEL) assays. In all six OSCC cell lines, cell growth was more inhibited by paclitaxel alone than combination therapy. Cisplatin-induced overexpression of FOXM1 showed the same trend only in cisplatin-resistant cell lines, indicating that it was associated with inhibition of paclitaxel-related apoptosis. In summary, these results suggest that, in three cisplatin-resistant cell lines, the combination of cisplatin and paclitaxel had an antagonistic effect, likely because cisplatin blocks paclitaxel-induced apoptosis. Cisplatin-induced FOXM1 overexpression may explain the failure of this combination.

## 1. Introduction

According to the GLOBOCAN 2018 report, oral cancer is estimated to have 354,864 (males 246,420; females 108,444) new cases and 177,384 (males 119,693; females 57,691) deaths worldwide in a single year [1]. Although early diagnosis is relatively easy because cancer cells can be distinguished with the naked eye, symptoms are often not noticed at an early stage because of their similarity to stomatitis [2]. Chemotherapy is classically used as a method of treating oral cancer [3].

Cisplatin, *cis*-diamminedichloridoplatinum (II), is known as an anticancer drug that inhibits the proliferation of cancer cells through the formation of intra-strand crosslinks with the purine bases in DNA [4,5]. It has been widely used to treat various cancers, including oral cancer, in cisplatin-based chemotherapy [6,7,8]. However, acquired resistance to cisplatin often impedes its therapeutic efficacy [6,9]. Combining cisplatin with other anticancer drugs may overcome this resistance and provide new treatment strategies for many cancers [4].

Paclitaxel, or taxol, is an antimicrotubule agent that binds to microtubules during cell division and induces apoptosis [4,10]. In some cisplatin-resistant cancer cell lines, cisplatin and paclitaxel combination chemotherapy was reported as a method of modulating cisplatin sensitivity [10,11]. However, the evaluation of cisplatin and paclitaxel combination therapy is divided, with evidence of both synergistic and antagonistic effects [10,11,12].

Mammalian transcription factor forkhead box M1 (FOXM1) is associated with cell proliferation and can be upregulated in cancer [13,14]. Specifically, these previous data show that overexpression of FOXM1 plays a role in cisplatin chemoresistance. More recently, FOXM1 had been shown to contribute to paclitaxel resistance by blocking mitosis-linked cell death in ovarian cancer cells [15].

We previously established three cisplatin-resistant human OSCC cell lines [16]. In the present study, we aimed to further investigate the effects of FOXM1 expression on these cells by applying cisplatin and paclitaxel in combination.

## 2. Results

### 2.1. OSCC Cell Line Growth is More Inhibited by Paclitaxel Alone than by the Combination of Paclitaxel and Cisplatin

First, we confirmed the effect of cisplatin and paclitaxel on cell growth of three parental cell lines (YD-8, YD-9, and YD-38) and three derived cisplatin-resistant cell lines (YD-8/CIS, YD-9/CIS, and YD-38/CIS) over 24 h and 48 h. An MTT assay was performed to determine whether the combination of cisplatin and paclitaxel increased cell growth inhibition compared to the effect of each alone. The IC_50_ values of cisplatin and paclitaxel in each cell line are summarized in Table 1. Figure 1A,B shows that cell growth was inhibited in a time-dependent manner by cisplatin treatment in parental cell lines, but there was little change in cisplatin-resistant cell lines. When paclitaxel was the treatment, cell growth of all six cell lines was significantly suppressed in a dose- and time-dependent manner. In both parental and cisplatin-resistant cell lines, cell growth was most inhibited by paclitaxel (0.1 μg/mL) alone. However, this was true only at the 24 h timepoint. After 48 h incubation, cisplatin with or without paclitaxel had a significant inhibitory effect on cell growth of the parental cell lines. We performed a combination index (CI) analysis to determine whether there was a synergistic effect of the combination treatment and found no synergism; on the contrary, the CI value was above 10 in all six cell lines, indicating very strong antagonism (Figure 1C and Table 2). These results confirmed that there was no synergistic effect between cisplatin and paclitaxel. Moreover, in cisplatin-resistant cell lines, there was a greater inhibition of cell growth with paclitaxel alone than with the combined treatment.

### 2.2. FOXM1 Expression is Upregulated by Cisplatin in Cisplatin-Resistant Cell Lines, but Downregulated by Cisplatin and Paclitaxel Combination Treatment 

Because FOXM1 overexpression has been reported in cisplatin-resistant cancer cells [13,14], we next used qPCR to investigate the expression of FOXM1 mRNA in both the parental and cisplatin-resistant cell lines after 6 h treatment with cisplatin and/or paclitaxel (Figure 2A). When treated with cisplatin alone, *FOXM1* was significantly upregulated in cisplatin-resistant cell lines, while the combination of cisplatin and paclitaxel had markedly lower *FOXM1* expression (2.9- vs 1.1-fold change for YD-8/CIS, 2.8- vs 1.1-fold change for YD-9/CIS, and 4.7- vs 1.0-fold change for YD-38/CIS cell lines). These changes in FOXM1 expression were confirmed by Western blot (Figure 2B,C). In the cisplatin-resistant cell lines, FOXM1 protein levels showed a trend similar to the change in gene expression. In the parental cell lines, neither FOXM1 mRNA nor protein was changed by cisplatin and/or paclitaxel treatment. Overall, these results show that in cisplatin-resistant cell lines, cisplatin increased FOXM1 expression, but the combination of cisplatin and paclitaxel attenuated this increase.

### 2.3. FOXM1 Protein Overexpression by Cisplatin Induces Apoptosis Reduction by Paclitaxel in Cisplatin-Resistant Cell Lines

Cisplatin was reported to inhibit apoptosis by paclitaxel, and FOXM1 regulated cisplatin sensitivity [12,14]. We assessed this in our cisplatin-resistant cell lines by Western blot, propidium iodide (PI) staining, and TdT dUTP nick end labeling (TUNEL) assays. In cisplatin-resistant cell lines, cisplatin increased the expression of FOXM1 protein and decreased the expression of cleaved-poly (ADP-ribose) polymerase (PARP) protein by paclitaxel (Figure 3A,B). The increased expression of FOXM1 caused by cisplatin treatment was decreased by treatment with forkhead domain inhibitory compound 6 (FDI-6), an inhibitor of FOXM1, and the amount of cleaved-PARP expression by paclitaxel was increased. We next examined apoptotic cell death. Figure 3C,D shows that, when cisplatin and paclitaxel were used in combination, sub-G_1_ phase population is greater in the presence of FDI-6 than in the absence of FDI-6 (YD-8/CIS, 8.2% vs. 5.8%; YD-9/CIS, 12.6% vs. 6.7%; YD-38/CIS, 15.2% vs. 10.7%). Similarly, the intracellular accumulation of terminal deoxynucleotidyl transferase (rTdT) with cisplatin and paclitaxel combination therapy was significantly higher with FDI-6 than without it (Figure 4A,B: YD-8/CIS, 12.9% vs. 7.2%; YD-9/CIS, 12.6% vs. 5.5%; YD-38/CIS, 20.4% vs. 13.2%). Overall, these data indicate that in the three cisplatin-resistant cell lines, cisplatin reduced the apoptosis caused by paclitaxel by causing overexpression of FOXM1. Thus, our data demonstrate that the mechanism by which cisplatin desensitizes the anticancer activity of paclitaxel in cisplatin-resistant cell lines may be increased via FOXM1 expression (Figure 4C).

## 3. Discussion

Combination chemotherapy with cisplatin and paclitaxel is currently used in several cancer treatments due to the different mechanisms of action of these two drugs, including cisplatin’s inhibition of DNA replication and paclitaxel’s inhibition of mitotic spindle formation [12,17,18]. However, some studies have claimed that this combination therapy is ineffective for cancer treatment [12,19]. Further, several studies in cancer cells and chemoresistant patients have shown that FOXM1 regulation can improve the efficacy of cisplatin and paclitaxel [14,20,21,22]. FOXM1 is an oncogenic transcription factor known to play an important role in anticancer drug resistance, and it promotes cancer progression [20,23,24,25]. In this study, we aimed to confirm the effect of cisplatin and paclitaxel combination therapy and the relationship to expression of FOXM1 in cisplatin-resistant cell lines and their parental cell lines.

We analyzed the association between cisplatin-induced FOXM1 overexpression and paclitaxel-related apoptosis in cisplatin-resistant cell lines. Our results showed that when parent and cisplatin-resistant OSCC cell lines were treated with cisplatin and paclitaxel in combination or alone, the highest cell growth inhibition was achieved with paclitaxel alone. Using a CI analysis, we confirmed that there is an antagonistic effect of cisplatin and paclitaxel co-treatment on OSCC cell lines. FOXM1 was previously reported that it correlates with chemoresistance in cancer cells and patients [13,14,21,26] and is upregulated in ovarian cancer cells after cisplatin treatment [14], and its overexpression in some cancers contributes to paclitaxel resistance [15,21,27]. Therefore, we investigated whether cisplatin induces FOXM1 mRNA and protein expression in OSCC cell lines. We found that cisplatin markedly induced FOXM1 mRNA and protein expression in cisplatin-resistant cell lines. However, in the non-resistant parental cell lines, FOXM1 protein expression increased slightly after cisplatin treatment in YD-8 and YD-9 cells but significantly increased in YD-38 cells. Next, we assessed the effect of a FOXM1 inhibitor (FDI-6) on paclitaxel-related apoptotic cell death in cisplatin-induced FOXM1-overexpressing cisplatin-resistant cells. We observed that cisplatin inhibited paclitaxel-associated apoptosis by increasing FOXM1 expression in cisplatin-resistant cell lines because apoptosis was restored by FDI-6, which reduces the expression of FOXM1 protein.

In conclusion, our results indicate that in cisplatin-resistant cell lines, paclitaxel alone has a better therapeutic effect than paclitaxel in combination with cisplatin or cisplatin alone. Our results further suggest that the overexpression of FOXM1 protein by cisplatin makes it difficult to overcome drug resistance to cisplatin and causes resistance to paclitaxel, which can potentially attenuate the effectiveness of combination chemotherapeutics on oral cancer. Based on these findings, we propose that patients may benefit considerably from paclitaxel monotherapy rather than a cisplatin–paclitaxel combination therapy as a second-line regimen when first-line cisplatin regimen has failed.

## 4. Materials and Methods

### 4.1. Reagents

Cisplatin (PubChem CID: 84691) was obtained from Choongwae Pharm. C. (Seoul, Korea) and dissolved in distilled water at 1 mg/mL. FDI-6 (PubChem CID: 329825840) was provided by Sigma (St. Louis, MO, USA) and dissolved in DMSO at 20 mM. The stock solutions were stored in aliquots at −20 °C until use. Paclitaxel (PubChem CID: 36314) was purchased from Hanmi Pharm. Co., Ltd. (Seoul, Korea), and it was used as an injection; the solvent is known as polyoxyl 35 castor oil and anhydrous alcohol. When cisplatin and paclitaxel were mixed, they were directly added to the cell culture media.

### 4.2. Cell Lines and Cell Cultures

The original OSCC cell lines (YD-8, YD-9, and YD-38) were purchased from the Korean Cell Line Bank (Seoul, Korea), and cisplatin-resistant cell lines (YD-8/CIS, YD-9/CIS, and YD-38/CIS) were derived from them using the methods described in our previous paper [16].

### 4.3. Cell Proliferation Assays

For cell proliferation assays, cells in suspension were seeded at 1 × 10^4^ cells per well in 96-well plates and then treated with cisplatin (0–10 μg/mL) and paclitaxel (0–0.1 μg/mL) the next day. After 24 or 48 h incubation, 0.5 mg/mL of MTT reagent (Sigma, St. Louis, MO, USA) was added into each well. The reagent was incubated for 2 h, then the medium was discarded. The remaining purple formazan crystals were dissolved in 100 μL of DMSO, and absorbance at 570 nm was measured using a SpectraMax Plus 384 microplate reader (Molecular Devices, Toronto, Canada).

### 4.4. Combination Index (CI) Calculation

Cells were treated with cisplatin (5, 8, or 10 μg/mL) and paclitaxel (0.1 μg/mL). The combination effects between cisplatin and paclitaxel were automatically simulated using CompuSyn 1.0 software (Paramus, NJ, USA). Values were defined as synergism, additive, and antagonism based on the CI ranges shown in Table 3 [28,29].

### 4.5. RNA Isolation and Real-Time qPCR

RNA was extracted from collected cells using an RNA-spin™ Total RNA Extraction Kit (iNtRON Biotechnology Inc., Seongnam, Korea) as previously described [16]. cDNA was directly synthesized from the RNA using High-Capacity cDNA Reverse Transcription Kits (Thermo Fisher Scientific, Waltham, MA, USA). The expression levels of *FOXM1* and *GAPDH* were analyzed by a qPCR ViiA7 Real-Time PCR System using GoTaq^®^ qPCR Master Mix (Promega, Madison, Wisconsin, USA). The following primers were used: *FOXM1*, 5′-AGCGACAGGTTAAGGTTGAG-3′ (forward) and 5′-GTGCTGTTGATGGCGAATTG-3′ (reverse) and *GAPDH*, 5′-AATCCCATCACCATCTTCCA-3′ (forward), and 5′-TGGACTCCACGACGTACTCA-3′ (reverse). The relative fold gene expression for each sample was calculated by the 2^−ΔΔCT^ method.

### 4.6. Western Blot Assay

We performed electrophoresis and blotting to investigate the expression levels of FOXM1, PARP, actin, and β-tubulin in the parent and resistant cell lines. Whole cell lysates (15 μg) were loaded onto an 8% SDS-PAGE gel and transferred to a PVDF membrane (Bio-rad, Hercules, CA, USA). After transfer, the membrane was incubated overnight at 4 °C with the following antibodies: anti-FOXM1 from Santa Cruz Biotechnology (Santa Cruz, CA, USA), anti-PARP purchased from CellSignaling Technology (Danvers, MA, USA), and antiactin and -α-tubulin obtained from Abcam (Cambridge, UK). Signals were detected using the EZ-western detection kit (Daeilab service Co. Ltd., Seoul, Korea) and analyzed by a ChemiDoc Touch imaging system (Bio-Rad, Hercules, CA, USA) and an automatic X-ray film processor JP-33 (JPI Healthcare Co., Ltd., Seoul, Korea).

### 4.7. Propidium Iodide (PI) Staining Assay

For the PI staining assay, cells were seeded at 1 × 10^6^ cells per dish in 100 mm culture dishes. The cells were pretreated with FDI-6 (5 μM) for 3 h and then treated with cisplatin (10 μg/mL) and paclitaxel (0.1 μg/mL). After 24 h incubation, the cells were harvested by centrifugation at 300× *g* for 5 min, washed twice in ice-cold phosphate-buffered saline (PBS), fixed in 70% ethanol, and stored at −20 °C for at least 24 h. Next, the cells were washed with ice-cold PBS and resuspended in 500 μL of PI/RNase Staining Buffer. Dead cells were measured by flow cytometry using a FACSCalibur (Becton-Dickinson, San Jose, CA, USA), and the sub-G_1_ population was analyzed using WinMDI 2.8 software (Joseph Trotter, La Jolla, CA, USA).

### 4.8. TUNEL Assay

Apoptosis was quantified using the DeadEnd™ Fluorometric TUNEL System (Promega, Madison, WI, USA) according to the manufacturer’s instructions. Apoptotic cells were measured by labeling with fluorescein isothiocyanate (FITC)/dUTP, using recombinant rTdT enzyme and PI. Briefly, cells were harvested by centrifugation (300× *g*) at −4 °C for 10 min. The cell pellets were resuspended in 0.5 mL of PBS and fixed by adding 5 mL of 1% paraformaldehyde for 20 min on ice. Next, the cells were washed twice in ice-cold PBS and permeabilized by adding 5 mL of 70% ice-cold ethanol at −4 °C overnight. The cells were washed twice in PBS, resuspended in 50 μL of rTdT incubation buffer, and incubated in a water bath for 1 h at 37 °C. To terminate the reaction, 1 mL of 20 mM EDTA was added to the cells. Finally, the cells were washed twice in 1 mL of a 0.1% Triton X-100 solution in PBS containing 5 mg/mL BSA, resuspended in 0.5 mL of PI solution, and incubated for 15 min at room temperature in the dark. Green fluorescence, which exists only in the nucleus of apoptotic cells, and red fluorescence were detected by flow cytometry using a FACSCalibur (Becton-Dickinson, San Jose, CA, USA). Data were analyzed using FlowJo 10.7 software (Becton-Dickinson, San Jose, CA, USA).

### 4.9. Statistical Analysis

All data were presented as the mean ± SD of at least three independent experiments, and statistical analyses were performed using Student’s t-test in Microsoft Excel (Microsoft Corporation, Redmond, WA, USA) and one-way ANOVA in GraphPad Prism 9.0 (GraphPad Software, San Diego, CA, USA). A *p* value of <0.05 was considered statistically significant.

## Figures and Tables

**Figure 1 ijms-21-08897-f001:**
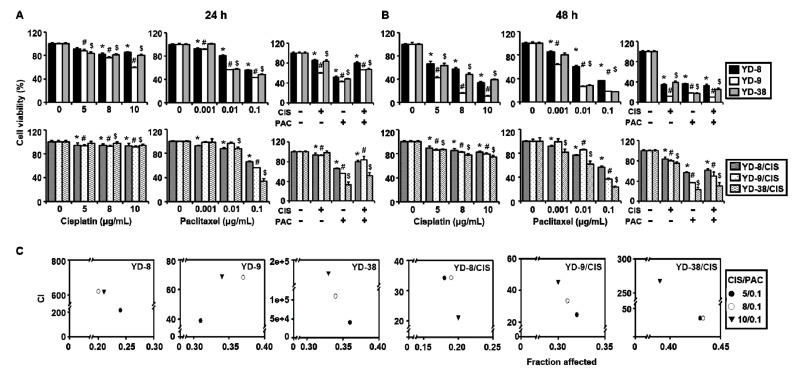
Cisplatin and paclitaxel used in combination had no synergistic effect in oral squamous cell carcinoma (OSCC) cancer cell lines. (**A**,**B**) Cells were treated with cisplatin (0–10 μg/mL) or paclitaxel (0–0.1 μg/mL) for 24 and 48 h. After 24 h, cell growth of these cancer cell lines was more inhibited by paclitaxel (0.1 μg/mL) alone than by cisplatin (10 μg/mL) alone or by combined treatment. The effect of cisplatin with or without paclitaxel on the viability of YD-8, YD-9, YD-38, YD-8/CIS, YD-9/CIS, and YD-38/CIS cells was determined by MTT assay (mean ± SD; *n* = 6). * *p* < 0.05 versus non-treated group in YD-8 or YD-8/CIS cells, # *p <* 0.05 versus non-treated group in YD-9 or YD-9/CIS cells, and $ *p* < 0.05 versus non-treated group in YD-38 or YD-38/CIS cells. (**C**) There was no synergistic effect. A fraction-affected versus CI plot (CIS/PAC) was determined using the Chou-Talalay and CompuSyn software.

**Figure 2 ijms-21-08897-f002:**
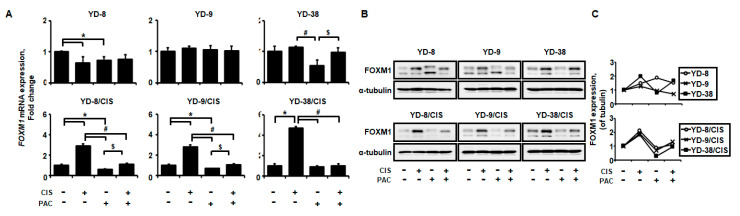
Cisplatin (10 μg/mL) incubation for 6 h induces overexpression of FOXM1 mRNA and protein in cisplatin-resistant cell lines, whereas combination treatment attenuates this increase. (**A**) The expression of FOXM1 mRNA from these six cell lines treated with cisplatin in the presence or absence of paclitaxel was determined by real-time qRT-PCR (mean ± SD; *n* = 3). * *p* < 0.05 versus non-treated group, # *p <* 0.05 versus only cisplatin-treated cells, and $ *p* < 0.05 versus only paclitaxel-treated cells. (**B**) The expressions of FOXM1 and α-tubulin in cell lysates from these six cell lines treated with cisplatin in the presence and absence of paclitaxel were determined by Western blot assay. (**C**) Quantification of Western blot results.

**Figure 3 ijms-21-08897-f003:**
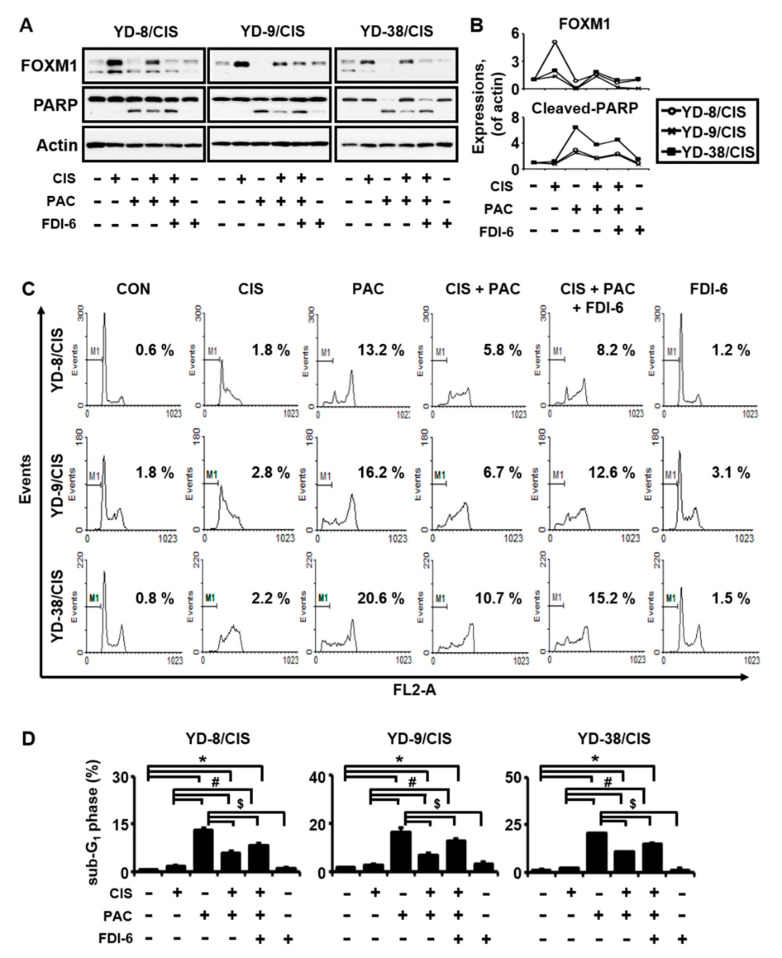
Cisplatin (10 μg/mL) weakens the chemical sensitivity of paclitaxel (0.1 μg/mL), whereas co-treatment with FDI-6, a FOXM1 inhibitor, restores it in cisplatin-resistant cell lines. (**A**) The expression of FOXM1, poly (ADP-ribose) polymerase (PARP), and actin in cell lysates from the cisplatin-resistant cell lines treated with cisplatin in the presence and absence of paclitaxel and FDI-6 were determined by Western blot assay. (**B**) Quantification of Western blot results. (**C**) The cytotoxic effect of cisplatin with or without paclitaxel in cisplatin-resistant cell lines was determined by a propidium iodide (PI) staining assay. (**D**) Quantitative results for panel C (mean ± SD; *n* = 3). * *p* < 0.05 versus non-treated group, # *p* < 0.05 versus only cisplatin-treated cells, and $ *p* < 0.05 versus only paclitaxel-treated cells.

**Figure 4 ijms-21-08897-f004:**
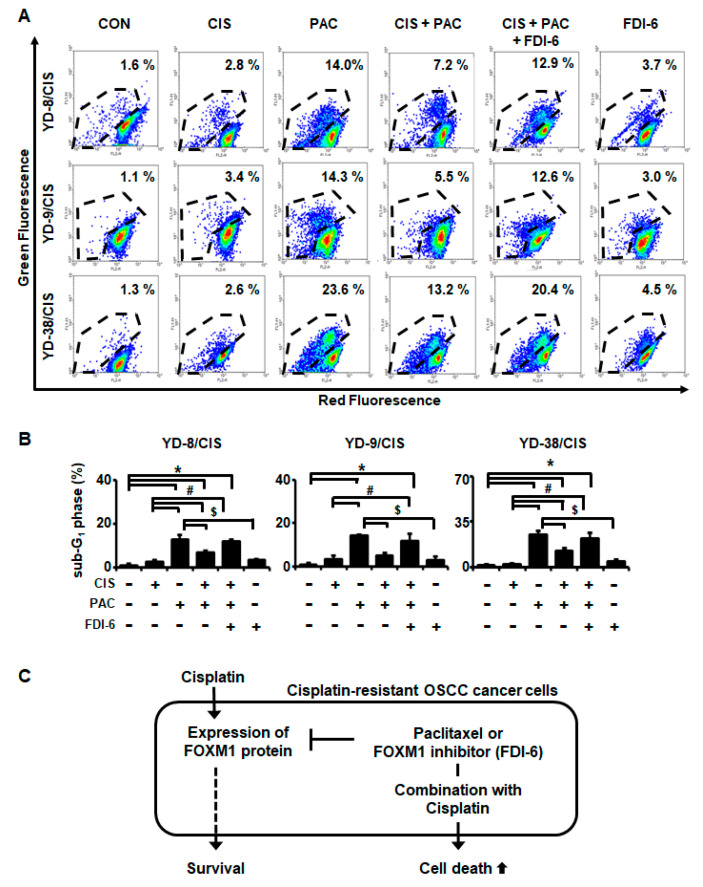
Treatment with FDI-6 in combination with cisplatin (10 μg/mL) and paclitaxel (0.10 μg/mL) for 24 h enhances apoptosis in cisplatin-resistant cell lines. (**A**) The fragmented DNA of apoptotic cells by each drug or combination treatment in cisplatin-resistant cell lines for 24 h was measured by a TUNEL assay. DNA strand breaks were measured by dual staining with rTdT enzyme (green fluorescence) and PI (red fluorescence). Data represent quantitative results for cells positive for rTdT enzyme in panel A of this figure. (**B**) Data represent quantitative results for panel A (mean ± SD; *n* = 3). * *p* < 0.05 versus non-treated group, # *p* < 0.05 versus only cisplatin-treated cells, and $ *p* < 0.05 versus only paclitaxel-treated cells. (**C**) Schematic diagram showing that cisplatin reduces paclitaxel-mediated apoptosis by overexpressing FOXM1 protein in cisplatin-resistant OSCC cancer cell lines.

**Table 1 ijms-21-08897-t001:** IC_50_ values (μg/mL) of cisplatin and paclitaxel in OSCC cells.

Cell Line	[Cisplatin]		[Paclitaxel]		Cell Line	[Cisplatin]		[Paclitaxel]	
	24 h	48 h	24 h	48 h		24 h	48 h	24 h	48 h
YD-8	>10	8.07 ± 0.24	>0.1	0.31 ± 0.06	YD-8/CIS	>10	>10	>0.1	>0.1
YD-9	>10	2.68 ± 0.26	0.06 ± 0.13	0.003 ± 0.08	YD-9/CIS	>10	>10	>0.1	0.06 ± 0.28
YD-38	>10	7.46 ± 0.08	0.04 ± 0.48	0.005 ± 0.12	YD-38/CIS	>10	>10	0.07 ± 0.09	0.02 ± 0.05

Each value represents the mean of at least three independent experiments (n = 6, mean ± SD). IC_50_, half maximal inhibitory concentration.

**Table 2 ijms-21-08897-t002:** CI analysis of cisplatin combined with paclitaxel used to treat OSCC cells.

Cell Line	[Cisplatin]	[Paclitaxel]	Fa	CI	Description	Cell Line	[Cisplatin]	[Paclitaxel]	Fa	CI	Description
YD-8	5 μg/mL	0.01 μg/mL	0.17	>10	Very strong antagonism	YD-8/CIS	5 μg/mL	0.01 μg/mL	0.20	2.2	Antagonism
	8 μg/mL		0.17	>10			8 μg/mL		0.16	5.9	Strong antagonism
	10 μg/mL		0.14	>10			10 μg/mL		0.18	3.6	Strong antagonism
	5 μg/mL	0.1 μg/mL	0.24	>10	Very strong antagonism		5 μg/mL	0.1 μg/mL	0.18	>10	Very strong antagonism
	8 μg/mL		0.20	>10			8 μg/mL		0.18	>10	
	10 μg/mL		0.21	>10			10 μg/mL		0.15	>10	
YD-9	5 μg/mL	0.01 μg/mL	0.20	>10	Very strong antagonism	YD-9/CIS	5 μg/mL	0.01 μg/mL	0.29	6.2	Strong antagonism
	8 μg/mL		0.30	1.7	Antagonism		8 μg/mL		0.29	6.2	Strong antagonism
	10 μg/mL		0.30	2.8	Antagonism		10 μg/mL		0.26	>10	Very strong antagonism
	5 μg/mL	0.1 μg/mL	0.31	>10	Very strong antagonism		5 μg/mL	0.1 μg/mL	0.32	>10	Very strong antagonism
	8 μg/mL		0.37	>10			8 μg/mL		0.31	>10	
	10 μg/mL		0.33	>10			10 μg/mL		0.30	>10	
YD-38	5 μg/mL	0.01 μg/mL	0.46	>10	Very strong antagonism	YD-38/CIS	5 μg/mL	0.01 μg/mL	0.45	1.3	Slight antagonism
	8 μg/mL		0.43	>10			8 μg/mL		0.42	5.4	Strong antagonism
	10 μg/mL		0.45	>10			10 μg/mL		0.40	>10	Very strong antagonism
	5 μg/mL	0.1 μg/mL	0.36	>10	Very strong antagonism		5 μg/mL	0.1 μg/mL	0.43	>10	Very strong antagonism
	8 μg/mL		0.34	>10			8 μg/mL		0.43	>10	
	10 μg/mL		0.41	>10			10 μg/mL		0.42	>10	

**Table 3 ijms-21-08897-t003:** CI value description.

Value Range	Description	Value Range	Description
<0.3	Strong synergism	1.10–1.20	Modetate antagonism
0.3–0.7	Synergism	1.20–1.45	Slight antagonism
0.7–0.85	Moderate synergism	1.45–3.3	Antagonism
0.85–0.9	Slight synergism	3.3–10	Strong antagonism
0.90–1.10	Nearly additive	>10	Very strong antagonism

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
