# Peer review of "Increased FOXM1 Expression by Cisplatin Inhibits Paclitaxel-Related Apoptosis in Cisplatin-Resistant Human Oral Squamous Cell Carcinoma (OSCC) Cell Lines"

_ijms, 2020, doi:10.3390/ijms21238897_

Round 1

Reviewer 1 Report

The article is written clearly and understandably. The authors deal with the problem of cisplatin and paclitaxel therapy. Is combined or sequential treatment with these drugs better and what about patients who show platinum resistance? This form of therapy is widely applicable, with relatively high toxicity often grades 3 and 4. If there is a marker of cisplatin resistance, this would greatly facilitate an early change in drug choice.
I think the results are very interesting. I hope it will encourage similar research on other types of tumors to get a more accurate picture. It should be noted that the experiment was conducted on cell culture, as clinical data show that combination therapy achieves a faster response, but with moderate to severe side effects.

Author Response

The article is written clearly and understandably. The authors deal with the problem of cisplatin and paclitaxel therapy. Is combined or sequential treatment with these drugs better and what about patients who show platinum resistance? This form of therapy is widely applicable, with relatively high toxicity often grades 3 and 4. If there is a marker of cisplatin resistance, this would greatly facilitate an early change in drug choice.
I think the results are very interesting. I hope it will encourage similar research on other types of tumors to get a more accurate picture. It should be noted that the experiment was conducted on cell culture, as clinical data show that combination therapy achieves a faster response, but with moderate to severe side effects.

Thanks for your good review comments, but the purpose of this study was to confirm the combination effect of cisplatin and paclitaxel in three cisplatin-resistant oral cancer cell lines constructed in our laboratory. In the future, as your advice, we will study whether other types of tumors show similar trends to provide a sufficient basis for this result.

Reviewer 2 Report

After reading the manuscript my major concerns are as follows:

  1. The authors must present the IC50 values for cisplatin (CIS) and paclitaxel (PAC) when used alone in all 10 cancer cell lines. The authors must confirm that the drugs have anti-proliferative effects in these cell lines. Otherwise, the interaction analysis based on the combination index makes no sense. There was no information about the concentration-response relationships for the anti-proliferative effects for CIS and PAC. At present, the readers do not know whether the drugs are really active or not in this study. Did the doses of the drugs have anti-cancer effects or not? The missing IC50 values for the tested drugs (CIS and PAC) are crucial for the whole interaction study.
  2. The authors have tested 3 concentrations of cisplatin (CIS = 5, 8 and 10 microgram/mL) with only one concentration of paclitaxel (PAC = 0.1 microgram/mL), claiming finally the antagonistic interaction between drugs. Please, indicate the drug-dose ratio for the tested two-drug mixture. Was the drug-dose ratio constant? It is crucial for the combination index analysis. Please, confirm that PAC 0.1 microgram/mL possesses the anti-proliferative effects in all cell lines tested. Additionally, assess the strength of the effects for PAC 0.1 microgram/mL. The same holds true for CIS.
  3. From a methodological point of view, please, indicate the pH of the drugs when they were used in combination. In the reviewer’s opinion, both drugs underwent pharmaceutic interaction before they were used together in cell lines. Was there any precipitation when CIS was added to PAC? Please, indicate the solvent for the mixture of both drugs (distilled water of DMSO?).
  4. Why all the experiments were conducted after 24 h of incubation except for the  combination index analysis (showing antagonistic interaction), which was performed after 48 hs of incubation? Why different incubation times were used in this study. Please, confirm this protocols by the respective references. Why not conducting experiments after 72hs of incubation? The anticancer effects are usually evaluated after 72 hours of incubation in in vitro studies. Please, explain this methodological change that certainly affected the results.
  5. Results presentation – Figures 2, 3, 4: please, explain number 3 (n=3) for the results presented as mean +/- SD. Was it in triplicate (as mentioned on page 8 of 10 line 250), or only 3 wells in the 96-well plate had the same concentration. This explanation is crucial for the interaction analysis. It is not clear, why the results presented in Figure 1 had n=6, whereas the results presented in Figures 2, 3 and 4 had n=3?
  6. Page 6 of 10 lines 185: Please, indicate the solvent for PAC and its stock solution, which was not mentioned in the paper.
  7. Table 1 on page 3 of 10: How to explain the fact that the combination index (CI) values for some combinations of CIS with PAC amounted to 184,580.0 and 110,299.0? Such a huge antagonism may results from precipitation of the drugs in mixture and their inactivation before they were used in the respective cell lines. Please, explain this phenomenon in light of molecular mechanisms of action. It is unusual that the CI values have such a huge values suggesting some methodological errors.
  8. Page 4 of 10: Lines 114-115: …”sub-G1 phase population is greater in the presence of FDI-6 than in the absence of FDI-6…” - what does it mean. Was the effect analyzed statistically or not? Please, indicate a statistical test used in this study to analyze the data.
  9. Figures 2, 3 and 4: The application of the Student’s t-test when comparing several values is inappropriate. Please, use one-way ANOVA to analyze 4 and 5 values for FOXM1 mRNA and FDI-6 (results present in Figures 2, 3 and 4). Besides, Figure 4B (on page 5 of 10) must contain description of drugs present in specific places (plus [+] and minus [-] make no sense without specific drug description).
  10. Page 8 of 10, line 252: Generally, a p-value should be established at 0.05 (5% of probability). It is not clear why the authors accepted a p-value less than 0.005 (p<0.005). Please, explain this exemption from a statistical viewpoint.

Minor concerns:

  1. Page 1, line 6 – the names of the authors: Who else should be mentioned as the co-authors of this study because at the end of this line “and” has an asterisk.
  2. Reference list: citation no 16 on page 9 of 10 – Please, insert the name of the journal that published the recent paper by the authors.
  3. Please, follow the rules and regulations of the International Journal of Molecular Sciences so as to properly arrange the citations.

Author Response

After reading the manuscript my major concerns are as follows:

  1. The authors must present the IC50 values for cisplatin (CIS) and paclitaxel (PAC) when used alone in all 10 cancer cell lines. The authors must confirm that the drugs have anti-proliferative effects in these cell lines. Otherwise, the interaction analysis based on the combination index makes no sense. There was no information about the concentration-response relationships for the anti-proliferative effects for CIS and PAC. At present, the readers do not know whether the drugs are really active or not in this study. Did the doses of the drugs have anti-cancer effects or not? The missing IC50 values for the tested drugs (CIS and PAC) are crucial for the whole interaction study.

Thanks for your good review comments. It was added to IC50 values in Table 1 and modified Figure 1 according to your comments.

  1. The authors have tested 3 concentrations of cisplatin (CIS = 5, 8 and 10 microgram/mL) with only one concentration of paclitaxel (PAC = 0.1 microgram/mL), claiming finally the antagonistic interaction between drugs. Please, indicate the drug-dose ratio for the tested two-drug mixture. Was the drug-dose ratio constant? It is crucial for the combination index analysis. Please, confirm that PAC 0.1 microgram/mL possesses the anti-proliferative effects in all cell lines tested. Additionally, assess the strength of the effects for PAC 0.1 microgram/mL. The same holds true for CIS.

Thanks for your comments. We actually experimented with paclitaxel concentrations of 0.01 and 0.1 with cisplatin, but no data were shown. However, we followed your advice and added the CI values obtained at the 0.01 concentration of paclitaxel to Table 2.

  1. From a methodological point of view, please, indicate the pH of the drugs when they were used in combination. In the reviewer’s opinion, both drugs underwent pharmaceutic interaction before they were used together in cell lines. Was there any precipitation when CIS was added to PAC? Please, indicate the solvent for the mixture of both drugs (distilled water of DMSO?).

Thanks for your comments. It was added to the solvent of paclitaxel in the Methods section according to your comments. When cisplatin and paclitaxel were mixed, they were directly added to the cell culture media.

  1. Why all the experiments were conducted after 24 h of incubation except for the combination index analysis (showing antagonistic interaction), which was performed after 48 hs of incubation? Why different incubation times were used in this study. Please, confirm this protocols by the respective references. Why not conducting experiments after 72hs of incubation? The anticancer effects are usually evaluated after 72 hours of incubation in in vitro studies. Please, explain this methodological change that certainly affected the results.

Thanks for your comments. Looking at the 48-hour data in our revised Figure 1, when cisplatin and paclitaxel were treated in combination, the cell growth was greatly inhibited even by each alone, making it difficult to confirm the combination effect. So, we conducted a follow-up experiment focusing on drug treatment data for 24 hours.

  1. Results presentation – Figures 2, 3, 4: please, explain number 3 (n=3) for the results presented as mean +/- SD. Was it in triplicate (as mentioned on page 8 of 10 line 250), or only 3 wells in the 96-well plate had the same concentration. This explanation is crucial for the interaction analysis. It is not clear, why the results presented in Figure 1 had n=6, whereas the results presented in Figures 2, 3 and 4 had n=3?

Thanks for your comments. We performed at least three independent experiments, MTT assay analyzed with 6 wells and other assay analyzed number 3 for each experiment.

  1. Page 6 of 10 lines 185: Please, indicate the solvent for PAC and its stock solution, which was not mentioned in the paper.

Thanks for your comments. It was added to the solvent of paclitaxel in the Methods section according to your comments. We used paclitaxel, which is currently being used as an injection, and the solvents are known as polyoxyl 35 castor oil and anhydrous alcohol.

  1. Table 1 on page 3 of 10: How to explain the fact that the combination index (CI) values for some combinations of CIS with PAC amounted to 184,580.0 and 110,299.0? Such a huge antagonism may results from precipitation of the drugs in mixture and their inactivation before they were used in the respective cell lines. Please, explain this phenomenon in light of molecular mechanisms of action. It is unusual that the CI values have such a huge values suggesting some methodological errors.

Thanks for your comment. We modified Table 2. The reason for the high numbers when calculating CI value using Compusyn software is probably because it is very antagonism. Therefore, referring to your advice, if the CI value is more than 10, we marked it as >10.

  1. Page 4 of 10: Lines 114-115: …”sub-G1 phase population is greater in the presence of FDI-6 than in the absence of FDI-6…” - what does it mean. Was the effect analyzed statistically or not? Please, indicate a statistical test used in this study to analyze the data.

Thanks for your comments. When cisplatin and paclitaxel are used in combination, FOXM1 expression by cisplatin increases and suppresses apoptosis by paclitaxel. However, the addition of the FOXM1 inhibitor (FDI-6) blocks this action, increasing sub-G1 phase population. Thus, these results explain that cisplatin attenuates apoptosis by paclitaxel. The statistical analysis results are shown in Figure 3D.

  1. Figures 2, 3 and 4: The application of the Student’s t-test when comparing several values is inappropriate. Please, use one-way ANOVA to analyze 4 and 5 values for FOXM1 mRNA and FDI-6 (results present in Figures 2, 3 and 4). Besides, Figure 4B (on page 5 of 10) must contain description of drugs present in specific places (plus [+] and minus [-] make no sense without specific drug description).

Thanks for your comment. We modified Figures 2-4. It was analyzed again with one-way ANOVA and added a description of Figure 4B according to your comments.

  1. Page 8 of 10, line 252: Generally, a p-value should be established at 0.05 (5% of probability). It is not clear why the authors accepted a p-value less than 0.005 (p<0.005). Please, explain this exemption from a statistical viewpoint.

Thanks for your comment. We used a p-value less than 0.005 to highlight significance. However, following your advice, we re-analyzed it with one-way ANOVA and corrected it less than 0.05.

Minor concerns:

  1. Page 1, line 6 – the names of the authors: Who else should be mentioned as the co-authors of this study because at the end of this line “and” has an asterisk.

Thanks for your comment. We noticed something wrong with your advices and corrected the manuscript.

  1. Reference list: citation no 16 on page 9 of 10 – Please, insert the name of the journal that published the recent paper by the authors.

Thanks for your comment, we have revised the reference.

  1. Please, follow the rules and regulations of the International Journal of Molecular Sciences so as to properly arrange the citations

We appreciate your comment. It was very helpful comments. We noticed something wrong with your advices and corrected the references section.

Round 2

Reviewer 2 Report

No further comments to the paper.